# Optional Vaccines in Children—Knowledge, Attitudes, and Practices in Romanian Parents

**DOI:** 10.3390/vaccines10030404

**Published:** 2022-03-07

**Authors:** Victor Daniel Miron, Andrei Răzvan Toma, Claudiu Filimon, Gabriela Bar, Mihai Craiu

**Affiliations:** 1Carol Davila University of Medicine and Pharmacy, 050474 Bucharest, Romania; claudiu.filimon@stud.umfcd.ro (C.F.); mihai.craiu@umfcd.ro (M.C.); 2National Institute for Mother and Child Health “Alessandrescu-Rusescu”, 020395 Bucharest, Romania; bargabriela@yahoo.com; 3Central Military Emergency Hospital Dr. Carol Davila, 010825 Bucharest, Romania; tomaandrei11@yahoo.com

**Keywords:** optional vaccine, children, knowledge, attitudes, practices, parents

## Abstract

Vaccination is one of the most useful medical interventions for controlling certain infectious diseases. The aim of current research is to identify some of the drivers of vaccine hesitancy or acceptance in a rather skeptical European population by addressing parental perception on optional vaccination (OV) perception. Novel tools, delivered by social media, were used in our research attempt. A validated questionnaire was distributed online among parents. Parental knowledge, attitudes and perceptions of OV were analyzed. The majority of parent respondents (55.1%) showed very good knowledge about vaccination and vaccine-preventable diseases, and 76.0% stated that they had given at least one optional vaccine to at least one of their children. The most common optional vaccine administered was the rotavirus vaccine. The level of knowledge appeared to be related to compliance with OV. Concurrently, the rate of vaccine acceptance in the NIP (National Immunization Program) was not correlated with the level of parental knowledge. In total, a high percentage of parents (77.6%) believed that OV can bring an additional health safety benefit to their children. This study shows the need to involve the medical community in a steady dialogue with parents about OV. Raising awareness by presenting clear and understandable information could be a game-changing intervention in mitigating the public health impact of OV-preventable diseases.

## 1. Introduction

Vaccination is one of the most important medical advances that has led to the eradication of smallpox [1] and a decrease in the prevalence of many infectious diseases in humans [2]. Many types of vaccines have been developed and upgraded over the years, and recent progress in vaccinology has led to major improvements in the control of infectious diseases. Each country, depending on its socioeconomic and epidemiological characteristics, has adopted its own vaccination schedule that is offered free of charge from birth. In addition, there are a number of vaccines for individual purchase, intended for additional protection. The latter are optional vaccines and are neither reimbursed nor compensated.

In Romania, the free vaccination schedule offered by the Ministry of Health (MoH) through the National Immunization Program (NIP) has undergone many changes over the last decades. It has been updated in accordance with the progress of vaccine development, the epidemiological evolution of vaccine-preventable diseases and the economic strength or the gross domestic product of the country. Currently, vaccines against 11 preventable diseases are included in the NIP and are administered in maternity wards and general practitioners offices: hepatitis B vaccine (HepB) and bacille Calmette Guerrin vaccine (BCG) are administered in maternity wards, followed by hexavalent vaccine (Hex) (diphtheria-tetanus-acellular pertussis-polio-*Haemophilus b*-hepatitis B) and pneumococcal conjugate vaccine (PCV13) administered at 2, 4 and 11 months; at the age of one year, the measles-mumps-rubella vaccine (MMR) is given, followed by a second dose at the age of five; at the age of six, the acellular diphtheria-tetanus-pertussis-inactivated polio vaccine (DTaP-IPV) is given and, at the age of 14, the acellular diphtheria-tetanus-pertussis booster (DTaP) [3]. In addition, at the parents’ request, all girls aged 11–18 can be vaccinated free of charge against human papilloma virus (HPV) [4]. Furthermore, a number of optional vaccines are available in pharmacies and can be administered to children: influenza vaccine (quadrivalent inactivated influenza vaccines—IIV4 and quadrivalent live attenuated influenza vaccine—LAIV4), rotavirus vaccine (RV), varicella vaccine (VAR), meningococcal vaccine (serogroups ACYW—MenACYW and serogroup B—MenB) and hepatitis A vaccine (HepA).

The vaccination coverage rate has decreased in recent years, mainly due to a worldwide antivaccination movement, which is also present in Romania [5]. Complacency, mistrust in authorities, procurement issues and other components of a malfunctioning system concurred in a constant decrease in vaccine coverage over the past 15 years in Romania [6]. As a consequence, our country has recently experienced a measles epidemic following a sharp decrease in MMR vaccination coverage [7]. Overall, the current vaccination coverage of children in Romania is about 86.3% for Hex, 85.3% for PCV13 and 86.2% for MMR [8]. At present, there are no systematized data on optional vaccination (OV) uptake in children from our country. The uptake of these vaccines depends on several socio-economic factors and on the involvement of physicians in informing parents about additional prophylaxis options [9,10].

Correct and comprehensible information for parents and debunking antivaccination fake news [11] is essential to increase vaccination rates, both free and optional. Social media currently plays an important role in disseminating information, and the quality of this information is paramount. In 2017, an online educational project entitled Spitalul Virtual pentru Copii^®^ (SVC, Virtual Children’s Hospital, https://www.facebook.com/drCraiuMihai/, accessed on 22 August 2021) was launched using the Facebook platform. Here, parents are provided with up-to-date medical information in easy-to-understand language. SVC has a wide reach among parents, with more than 246,000 users following this page so far. Free or optional vaccination has been addressed in this virtual space through numerous posts and discussions. In 2018, SVC won first prize in the Vaccines Today Communication Challenge [12].

Considering that data on OV in Romania is scarce and unstructured, and that parental perceptions on this topic have not been truly quantified, we conducted a questionnaire-based survey among parents interested in online medical education to assess their knowledge, perceptions and attitudes about OV in children.

## 2. Materials and Methods

We conducted a cross-sectional study by administering an online questionnaire with 29 questions (Appendix A) and collected general data on respondents and assessed their knowledge concerning vaccines and vaccine-preventable diseases, as well as their perceptions and attitudes towards OV.

The validity of the questionnaire was evaluated by piloting it on a subset of 20 parents who provided rationalized feedback on the consistency, relevance and acceptability of the questions and answers. This validation process was carried out between 10 and 13 August 2021. Their comments were taken into account when preparing the final version of the questionnaire, but their responses were not included in the final analysis.

After validation, the questionnaire was distributed online to parents in Google Forms format using the SVC Facebook platform. At the time the questionnaire was distributed, SVC had 218,309 likes, and 231,743 users were following the activity of this virtual space. The questionnaire was kept open for 7 days, from 16 August 2021 10:40 a.m. to 22 August 2021 1:10 p.m., and was broadcast through one post on SVC. Subsequently, by the time the survey closed, the post had been shared 52 times; it had received 624 reactions and gathered 221 comments. A total of 6205 visits were recorded to the post, of which 3163 were to the study link.

The responses were transferred to Microsoft Office Excel (Microsoft, Redmond, Washington), and the database was processed. Participants who incorrectly entered personal or children’s age data and participants who gave conflicting answers to questions assessing the same parameters were excluded from the analysis.

### 2.1. Knowledge Score and Knowledge Rating

The level of knowledge about vaccination and vaccine-preventable diseases was quantified in a “knowledge score” (KS) and “knowledge rating” (KR). This assessment was based on 3 questions: “Which of the following childhood diseases can be prevented by vaccination?”, “Which of the following vaccines are part of the free vaccination programme?” and “Until now, have you heard of any additional vaccines that can be given to your child/children?” (Appendix A). For the first two questions, respondents could choose more than one answer from a predefined list. For each correct answer, 1 point was allocated; for each wrong answer, 0.5 points were deducted. For the last question, 2 points were given for the answer “Yes” and 0 points for the answer “No”. The maximum knowledge score that could be obtained was 12 points. Knowledge ratings were calculated based on the knowledge score and classified as “very good”, “good”, “sufficient” and “insufficient” as follows: 10–12 points: “very good”, 7–9.5 points: “good”, 4.5–6.5 points: “sufficient” and less than or equal to 4 points: “insufficient”.

### 2.2. Statistical Analysis

The statistical analysis was performed with IBM SPSS Statistics for Windows, version 25 (IBM Corp., Armonk, NY, USA). For continuous variables with parametric distribution, we presented the mean values and the standard deviation (SD), and for continuous variables with non-Gaussian distribution, we presented the median and the interquartile range (IQR), while the differences between groups were analyzed with the Mann–Whitney *U* test and the Kruskal–Wallis H test. Effect size for the two tests was calculated as described in the literature [13]. For categorical variables, the frequencies and percentages were reported, and we used the chi-square test to calculate the odds ratio (OR) and its 95% confidence interval (95% CI). A *p*-value < 0.05 was considered statistically significant.

### 2.3. Ethical Approval

The study was approved by the Research Ethics Committee of the National Institute for Mother and Child Health “Alessandrescu-Rusescu”, Bucharest, with registration number 15243/2021. Consent to participate was obtained online as an initial part of the questionnaire. None of the data collected made it possible to identify respondents later.

## 3. Results

### 3.1. General Characteristics of Respondents

Out of a total of 2628 responses registered, 2550 were validated and included in our analysis. The majority of respondents were female (95.4%, *n* = 2432), lived in urban areas (88.9%, *n* = 2268), had higher education (90.1%, *n* = 2297) or had only one child (54.8%, *n* = 1397) (Table 1). The mean age ± SD of parent respondents was 36.9 ± 5.3 years (range: 22–54 years) (Table 1). Overall, the median age of the first child was 5 years (IQR: 3–9) (Table 1). In families with more than one child, the difference between the median ages of the children was about 1 year. Education level was not associated with the number of children, but rural respondents more frequently had two (50.7% vs. 39.9%, *p* < 0.001, χ^2^ = 12.2, OR = 1.6, 95% CI: 1.2–2.0) or three children (6.4% vs. 3.3%, *p* = 0.009, χ^2^ = 6.8, OR = 2.0, 95% CI: 1.2–3.4) compared to urban respondents, who more frequently had one child (56.4% vs. 42.2%, *p* < 0.001, χ^2^ = 20.4, OR = 1.7, 95% CI: 1.4–2.3).

### 3.2. Level of Knowledge

More than half of the parents (55.1%, *n* = 1406) had a “very good” level of knowledge about vaccination and vaccine-preventable diseases, and 35.7% (*n* = 910) had a “good” level. A total of 237 parents scored the maximum of 12 points. The median KS was 10 points (IQR: 8.5–11). Female respondents (*p* < 0.001) and those with higher education (*p* < 0.001) showed a higher level of knowledge. KS and KR were analyzed according to respondent characteristics and are detailed in Table 2.

### 3.3. Information and Communication

The main source of information on vaccinations, as reported by the respondents, was the pediatrician (78.7%, *n* = 2006) and/or the general practitioner (GP) (67.2%, *n* = 1713). However, online information (50.4%, *n* = 1284) or from other parents (31.6%, *n* = 805) were often used by respondents (Figure 1).

Almost all respondents (96.9%, *n* = 2472) had discussed at least once with their GP and/or pediatrician about the vaccines in the standard vaccination schedule offered free of charge by the MoH (Figure 2). By comparison, only 92.2% of parents (*n* = 2349/2548; two parents with missing data for this variable) (*p* < 0.001, χ^2^ = 55.9, OR = 2.6, 95% CI: 2.0–3.3) had discussed with their GP and/or pediatrician about OV. Overall, the pediatricians appeared to be more frequently involved in discussions about OV compared to the GPs (*p* < 0.001, χ^2^ = 12.3, OR = 1.3, 95% CI: 1.1–1.4) (Figure 2). A percentage of 97.7% (*n* = 2488/2547; three missing responses) considered that more active involvement of physicians in discussing free or OV is necessary. The median KS was higher for this category (10 (IQR: 8.5–11)) compared to parents who did not consider involving the doctor in vaccination-related decisions (9 (IQR: 8–10.5), *p* = 0.008, z = −2.636, r = 0.05). Parent sex, age, educational level, number of children or background were not associated with respondents` opinion of physician involvement in vaccination discussions (*p* > 0.05 for each).

### 3.4. Free Vaccination

Most of the respondents` children (91.6%, *n* = 2335) had been fully vaccinated for their respective ages, in accordance with the NIP; 7.3% (*n* = 185) had been partially vaccinated, 27 parents (1.0%) stated that none of their children had received any vaccine from the standard schedule and three parents did not know the vaccination status of their child. The main reasons for not vaccinating or partially vaccinating children included fear of adverse reactions (26.6%, *n* = 56/211; one parent did not answer this question), parents’ own decision to postpone vaccination (21.8%, *n* = 46/211) and a recommendation to avoid a particular vaccine from the GP (4.7%, *n* = 10/211) or the pediatrician (7.1%, *n* = 15/211).

We found no association between standard vaccination uptake and overall patient characteristics (*p* > 0.05 for all comparisons), KS (*p* = 0.081) and KR (*p* = 0.104).

An analysis of the 10 and 15 cases, respectively, in which the GP/pediatrician did not recommend one or more vaccines from NIP revealed that, in four cases, the doctor recommended taking none of the vaccines and, in 15 cases, MMR and, in four cases, DTaP at the age of 6 years.

### 3.5. Optional Vaccination

A percentage of 76.0% (*n* = 1937) of the responding parents stated that they had given at least one optional vaccine to at least one of their children. Among the reasons provided by parents who did not give any optional vaccine (24.0%, *n* = 613), the fear of possible adverse effects (31.2%, *n* = 191) ranked first. High costs (25.3%, *n* = 155) and a lack of clear and sufficient information on optional vaccines (22.3%, *n* = 137) were also cited by parents as reasons for not vaccinating. A total of 67 parents (10.9%) stated that they did not give any additional vaccines to their children based on the recommendation from their GP/pediatrician (Figure 3).

Parents’ choice of OV for their children appeared to be related to their level of knowledge about vaccination and vaccine-preventable diseases. Specifically, parents who administered at least one optional vaccine to their children had a significantly higher median KS compared to parents who did not use optional vaccines (10 points (IQR: 9–11.5) vs. 9 points (IQR: 7.5–10.5), *p* < 0.001, U = 403,747.0, z = −11.1, r = 0.22). This observation was also valid in the KR analysis in relation to vaccination choice (Table 3). The other general characteristics were not associated with OV choice.

According to the responses of the 1937 parents who had administered at least one additional vaccine to at least one of their children, this resulted in a total of 2881 children vaccinated (1067 parents with one child, 801 parents with two children, 64 parents with three children and 5 parents with four children), totaling 6068 additional vaccines administered (Table 4). The median ages of children vaccinated were as follows: first child—5 years (IQR: 3–9); second child—3 years (IQR: 1–7); third child—3 years (IQR: 1–5) and fourth child—0.7, 1, 2, 4 and 6 years. The most commonly administered optional vaccine was RV (28.1%, *n* = 1708), followed by influenza vaccine (19.2%, *n* = 1164 for IIV and 7.7%, *n* = 469 for LAIV4) and VAR (20.9%, *n* = 1267) (Table 4).

A high percentage of parents (77.6%, *n* = 1979) believed that OV brings an additional health safety benefit to their children. Therefore, 92.8% (*n* = 2366) of respondents believed that some of the optional vaccines should be included in the standard, free vaccination schedule. The following vaccines were among the most desired by parents to be included in the NIP: RV (67.9%, *n* = 1731), MenACYW (63.4%, *n* = 1616) and MenB (60.6%, *n* = 1546) (Figure 4).

We assessed which diseases preventable by OV are generating significant parental anxiety related to potential infection with the respective strain in their children. Meningococcal infection appeared to cause the greatest fear among parents (88.3%, *n* = 2251), followed by rotavirus infection (46.7%, *n* = 1191) and hepatitis A (42.4%, *n* = 1081) (Figure 5).

## 4. Discussion

In its latest report on immunization coverage, the World Health Organization (WHO) pointed out that, in 2020, the number of unvaccinated children increased by 3.4 million compared to 2019, causing the overall coverage to decrease by three percent, from 86% to 83% [14]. Childhood immunization coverage rates differ from country to country, even between regions within the same country, and parental reasons for vaccine uptake are highly heterogeneous, as shown in a systematic review by Dyda et al. [15] Romania follows the global trend and is experiencing a decline in NIP vaccination coverage [8]. Therefore, it is necessary to quantify the extent of the problem in order to identify measures to bring vaccination back to an upward curve. In addition to monitoring the administration of vaccines offered free by MoH, it is also crucial to analyze and understand the attitudes and perceptions of parents about OV and the diseases prevented by these vaccines. Our online survey targeted parents who are likely interested in validated medical information and provided a relevant snapshot of the vaccinal perspective of Romanian parents actively involved in their children’s health.

The internet and, especially, the use of social networks are an integral part of everyday life [16]. The “anti-vaxx” movement is very active online and has been a major contribution to the increase in vaccine hesitancy and refusal [11,17]. Potential measures that could address this issue include active debunking and fact-checking of misinformation in the same places where fake news circulate in social media.

We can provide validated medical information, presented in an apprehensible way [16,17]. SVC, among others, is such a space dedicated to correct information and parental counselling [18]. Topics relevant to parents are regularly discussed here, including issues related to vaccination or vaccine-preventable diseases. The large number of followers, from all over Romania, allowed us to apply this questionnaire in SVC in order to assess the magnitude of the problem.

A total of 2550 responses were validated and included in the analysis, and the majority of respondents were female. Mothers are known to be more actively involved in preventive measures and children’s health [19,20]. Jung M. [20] showed that better knowledge and understanding of vaccines and vaccine-preventable diseases among mothers can lead to increased vaccine coverage in the pediatric population. Most of the respondents in our study lived in cities. This may be a limitation of our study, as urban residence may play a role in positive perceptions of vaccination [21]. However, overall, in Romania, more than 56% of the population lives in urban areas [22]. Furthermore, the rural population is generally older, with an average of 42.2 years [23], while, in our study, the average age of respondents was 36.9 years.

The majority of parents demonstrated “good” and “very good” knowledge of vaccination and vaccine-preventable diseases. This correlates with the target group in which the questionnaire was distributed. We identified female parent sex and higher educational level as being associated with higher levels of knowledge, but data analysis identified a small effect size in both situations. This statistical finding is important to reduce the effect of a possible selection bias, as 90.1% of our respondents had higher education.

The main source of vaccine information for parents in our study was the doctor (pediatrician or GP), followed by the online environment. It is essential that physicians are updated on the efficacy and safety of vaccines and serve as advocates for their timely administration. The media and the internet have unfortunately focused aggressively on the controversies surrounding immunizations, often with nonscientific information. This attitude has led to increased parental anxiety, confusion and sometimes refusal to vaccinate [11,17]. Vaccine-hesitant parents are a growing group [24,25], and many of the measures we need to take need to be addressed towards them [25]. A key role is played by primary care physicians (GPs), who need to be prepared to talk to parents about the benefits and risks associated with vaccinations [25]. After all, history has shown that, for the vaccines currently in use, the risk is by far outweighed by the benefits of vaccination, both individually and at the macrosocial level. At the same time, it is important to improve the quality of the information provided to parents regarding vaccinations [26], as many of those surveyed in this study were of the opinion that the GP and/or pediatrician should be actively involved in discussions with parents regarding OV.

The uptake of free vaccines from the NIP among the respondents’ children was slightly higher than reported by the National Institute of Public Health in Romania (91% vs. 86%) [8], explained by the characteristics of the population studied. We ranked the main reasons for non-vaccination as follows: fear of possible adverse reactions and hesitancy. These data reinforce what was said above about the need to strengthen doctors` communication with the group of hesitant and insufficiently informed parents.

When addressing the topic of OV, about three quarters of parents said they had given at least one optional vaccine to at least one of their children. The level of knowledge about vaccination and vaccine-preventable diseases appeared to be related to the practice of optional vaccination; parents who administered at least one optional vaccine to their children had significantly higher KS and KR compared to those who did not give any optional vaccine. Concurrently, the rate of vaccine administration in the NIP does not correlate with the knowledge level. We have shown that uptake is dependent on a high level of knowledge about vaccines and vaccine-preventable diseases. Therefore, we believe that the main issue with OV in our country is the lack of correct, validated, parent-friendly information on “what exactly are optional vaccines?”; “what diseases do they protect us from?”; “what are the benefits and risks of optional vaccination?” and “how, where and when can they be given?”. Involving the medical community in answering these questions can be a game-changing intervention in mitigating OV-preventable diseases. Other issues raised by parents as an impediment to getting an optional vaccine need to be addressed, namely: high costs or availability in pharmacies.

A worrisome aspect highlighted by parents is the non-recommendation of vaccines by doctors themselves, with 15 cases for NIP vaccines and 67 cases for optional vaccines. This draws attention to the need for continuous medical education among doctors. It is not only parents who need clear information but also doctors. Anderson E. [27] pointed out that poorly trained medical staff can be a barrier to immunizing children. Furthermore, we identified recent, worrying data regarding Romanian medical students: only 90.7% of them would recommend giving an additional vaccine to a child or adult, and only 32.6% had received, for example, the influenza vaccine in the 2020/21 season [28].

Additionally, as shown in Table 4, there is a trend towards decreasing optional vaccinations for subsequent children. Thus, in families with more children, the uptake of optional vaccines is inversely proportional to the number of children. It is therefore important to target families with more than one child with clear and targeted actions in order to ensure an adequate optional vaccination rate.

The most common optional vaccine administered by parents in our study was the RV. Rotavirus infection is recognized as a major cause of gastroenteritis in infants and young children around the world and is associated with increased morbidity and mortality. In Romania, the incidence rate reaches 18.2% among children tested and hospitalized for acute diarrheal disease [6]. The fairly high rate of administration of this vaccine is most likely due to the fact that this vaccine is administered in the first 6 months of life when parents show increased interest towards protecting their infant. In addition, oral administration of this vaccine is a perceived advantage. In some countries, the rotavirus vaccine is included in the NIP, and the benefits have not been slow to appear. For example, in England, a significant reduction in direct healthcare costs has been reported in the first year after the introduction of RV [29]. Similarly, in Norway, an 86% decrease in cases admitted with rotavirus gastroenteritis in children <5 years in 2016 compared to 2014 to 2015 was observed [30,31]. RV ranks first in the wishes of parent respondents to be included in the free vaccination regimen. This perception is driven by frequent exposure to RV information and by fear of the disease, documented in 46.7% of respondents. A cost-effectiveness assessment of the introduction of this vaccine in the NIP is imperative.

Influenza prophylaxis of children by vaccination was reported in 26.9% of cases (19.2% IIV4, 7.2% LAIV4). The fairly high uptake of this optional vaccine is most likely due to the fact that influenza is a seasonally epidemic disease that is much talked about both in the media and social media every year. We should not forget that the fear of a double epidemic, influenza–COVID-19, has determined in 2020/21 the rapid exhaustion of influenza vaccine stocks in pharmacies in our country [32]. Influenza remains a seasonal disease that places a great burden on the pediatric population, associated with an increased hospitalization rate [33]. However, given that only 16.5% of respondents said they were afraid their child will acquire influenza, active effort is needed to increase the influenza vaccination rate each year [34].

Varicella vaccination ranked fourth in parents’ preferences in the current study. The incidence of varicella in Romania is high, at about 163 cases per 100,000 people [6], and the current vaccine coverage is not sufficient to ensure control of the disease. In addition, there is a popular belief in Romania that “chickenpox is just a childhood disease”, which is reflected by only 25.1% of parents expressing concern of disease regarding chickenpox. This is where medical education intervention is required and where parents` awareness of the disease needs to be raised. In a survey of Romanian parents who were hesitant to vaccinate, the VAR generated the greatest reluctance [35]. Most likely, their attitudes are fueled by the live-attenuated nature of the vaccine and by the lack of knowledge about VZV infection and its potential severity or the occurrence of latency and reactivation. In our study, the varicella vaccine ranked fourth among parents` preferences to be introduced in the NIP.

Meningococcal vaccine coverage is low at only 9.9% for ACYW serogroups and 6.9% for serogroup B. There could be at least three causes for this low rate: low incidence of meningococcal disease [6], high purchase costs and unavailability in pharmacies. However, parents are seriously concerned by meningococcal infection and want, in similar proportions, the introduction of both meningococcal vaccines in the NIP. This fear is justified, given that meningococcal disease can have a severe course potentially leading to death [36]. Meningococcal vaccines could be presented as OV to all parents to further reduce the burden of this disease.

The lowest vaccination rate in our study was for HepA. However, 42.4% of parents feared that their child would get hepatitis A, and a similar percentage wanted to introduce this vaccine in the NIP. Are there any potential explanations for this contradiction? Most probably because, in the Romanian parent`s perception, there is a confusion with hepatitis B or C. The mere diagnosis of hepatitis causes panic and worry. In Romania, about 5000 cases of hepatitis A were reported in 2018 [6], but many cases may go unnoticed [37]. There are indeed cases with fulminant evolution [38], and HepA vaccination is useful to reduce the burden of the disease. In their discussions with the parents, doctors should clarify the differences between the types of hepatitis A, B and C and to explain the prophylactic measures that can stop the transmission of the hepatitis A virus, including OV.

In addition to the above, this study comes with its own limitations and strengths. A potential selection bias towards respondents more actively engaged in health-seeking behaviors could be presumed, making our results less generalizable for underserved groups such as the rural population and for parents with lower education levels. However, this is, to the best of our knowledge, the first study to simultaneously address the knowledge, attitudes and practices of parents regarding routine and optional vaccinations in Romania and to highlight the main factors driving parents’ decisions whether to accept or refuse vaccinations. Our study’s results can thus lay the foundation for designing educational campaigns targeting GPs, pediatricians and parents, with the aim of mitigating false perceptions and educating on evidence-based pro-health decisions.

## 5. Conclusions

We have shown that there is a direct correlation between the level of knowledge about vaccination and vaccine-preventable diseases and the uptake of optional vaccines. Active involvement of the medical community in a constant dialogue with parents is paramount to increasing the awareness of OV and vaccine-preventable diseases. The way this information is conveyed should be tailored to each individual case. Vaccination-hesitant parents should be targeted first in order to increase vaccination rates. Social media can be a useful channel to present clear and easy-to-understand data.

## Figures and Tables

**Figure 1 vaccines-10-00404-f001:**
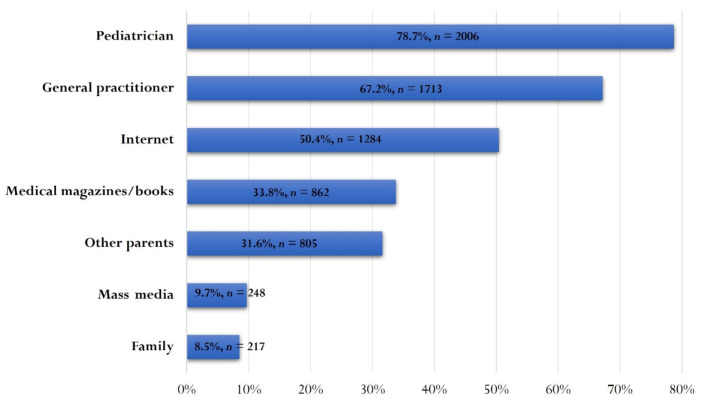
Parents’ sources of information on vaccinations.

**Figure 2 vaccines-10-00404-f002:**
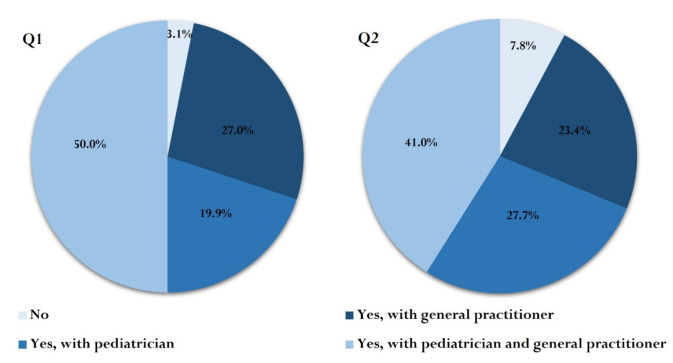
Communicating with your GP or pediatrician about vaccination. Q1. Have you ever discussed with your family doctor/pediatrician about vaccines in the free regimen for your child/children? Q2. Have you ever discussed with your family doctor/pediatrician about optional vaccines that can be given to your child/children?

**Figure 3 vaccines-10-00404-f003:**
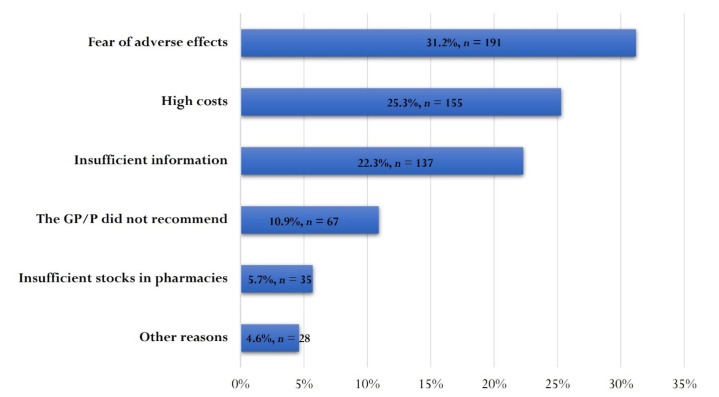
Parental reasons for not using OV in their children. GP—general practitioner; P—pediatrician.

**Figure 4 vaccines-10-00404-f004:**
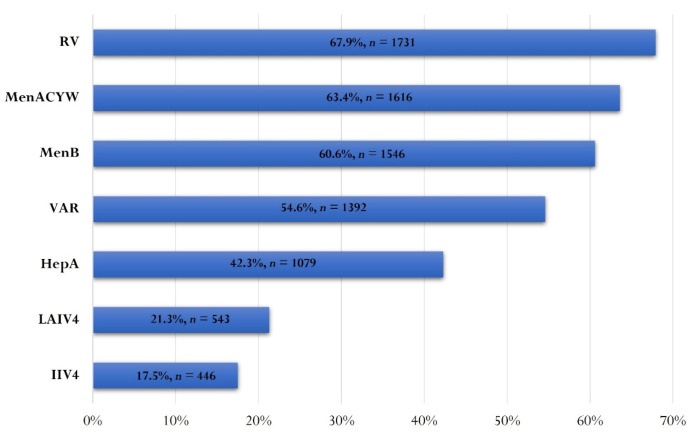
Optional vaccines that parents want to be included in the NIP. RV—rotaviral vaccine, MenACYW—meningococcal vaccine serogroups ACYW, MenB—meningococcal vaccine serogroup B, VAR—varicella vaccine, HepA—hepatitis A vaccine, LAIV4—quadrivalent live attenuated influenza vaccine and IIV4—quadrivalent inactivated influenza vaccine.

**Figure 5 vaccines-10-00404-f005:**
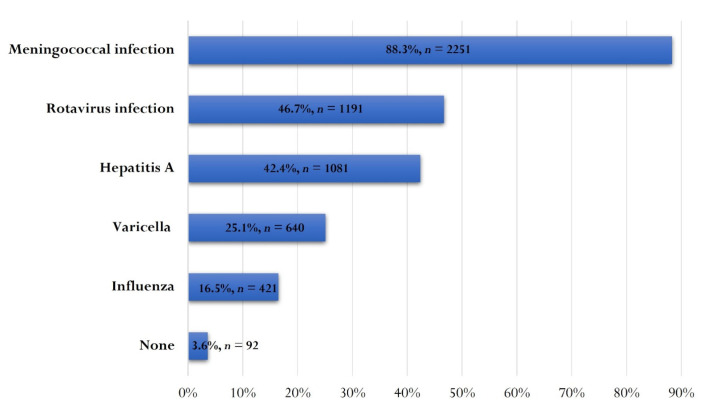
Diseases preventable by optional vaccination that parents fear one of their children will get sick from.

**Table 1 vaccines-10-00404-t001:** Demographic characteristics of respondents.

Characteristic	Number of Respondents, *n*(%)
Gender	
female	2432 (95.4)
male	118 (4.6)
Age (mean in years, ±SD)	
all respondents	36.9 ± 5.3
male	38.4 ± 5.2
female	36.8 ± 5.3
Residence	
urban	2268 (88.9)
rural	282 (11.1)
Educational level	
unfinished general school/without studies	2 (0.1)
completed general school	6 (0.2)
vocational school	10 (0.4)
high school	146 (5.7)
post-high school	89 (3.5)
university studies	2297 (90.1)
Number of children	
1 child	1398 (54.8)
2 children	1047 (41.1)
3 children	93 (3.6)
4 children	11 (0.4)
5 children	1 (0.05)
Median age of children (median in years, IQR)	
first child	5 (3–9)
second child	4 (2–7)
third child	3 (1–5.3)
fourth child	1 (0.3–4)

**Table 2 vaccines-10-00404-t002:** Knowledge score and knowledge rating by respondent characteristics.

Characteristic	Knowledge Score, Median (IQR)	Statistical Analysis	Knowledge Rating, *n*(%)				Statistical Analysis
			Very good	Good	Sufficient	Insufficient	
All participants (*n* = 2550)	10 (8.5–11)	NA	1406 (55.1)	910 (35.7)	211 (8.3)	23 (0.9)	NA
Gender							
female (*n* = 2432)	10 (8.5–11)	*p* < 0.001, U = 107,686.5, z = −4.604, r = 0.09	1359 (55.9)	866 (35.6)	190 (7.8)	17 (0.7)	*p* < 0.001, χ^2^(12) = 42.904, V = 0.130
male (*n* = 118)	8.5 (7–10.5)		47 (39.8)	44 (37.3)	21 (17.8)	6 (5.1)	
Age group							
under 30 years (*n* = 152)	10 (8–11)	*p* = 0.049, H(2) = 6.012, d = 0.079	78 (51.3)	55 (36.2)	18 (11.8)	1 (0.7)	*p* = 0.110, χ^2^(3) = 10.367
30–39 years (*n* = 1649)	10 (8.5–11)		932 (56.5)	586 (35.5)	117 (7.1)	14 (8.4)	
40 years and over (*n* = 749)	10 (8.5–11)		396 (52.9)	269 (35.9)	76 (10.1)	8 (1.1)	
Residence							
urban (*n* = 2268)	10 (8.5–11)	*p* = 0.187	1256 (55.4)	807 (35.5)	185 (8.2)	20 (0.9)	*p* = 0.870, χ^2^(3) = 0.715
rural (*n* = 282)	10 (8–11)		150 (53.2)	103 (36.5)	26 (9.2)	3 (1.1)	
Educational level							
unfinished general school (*n* = 2)	NA	*p* < 0.001, H(4) = 76.221, d = 0.342	2 (100)	0	0	0	*p* < 0.001, χ^2^(12) = 97.609, V = 0.113
completed general school (*n* = 6)	8.5 (7.9–9.5)		0	6 (100)	0	0	
vocational school (*n* = 10)	9 (5.9–10)		4 (40.0)	3 (30.0)	3 (30.0)	0	
high school (*n* = 146)	8.3 (6.5–10)		46 (31.5)	60 (41.1)	35 (24.0)	5 (3.4)	
post-high school (*n* = 89)	9.5 (8.5–11)		43 (4.8)	36 (40.4)	10 (11.2)	0	
university studies (*n* = 2297)	10 (8.5–11)		1311 (57.1)	805 (35.0)	163 (7.1)	18 (0.8)	
Number of children							
1 child (*n* = 1398)	10 (8.5–11)	*p* = 0.189	745 (53.3)	530 (37.9)	115 (8.2)	8 (0.6)	*p* = 0.038, χ^2^(12) = 21.975, V = 0.054
2 children (*n* = 1047)	10 (8.5–11)		601 (57.4)	344 (32.9)	90 (8.6)	12 (1.1)	
3 children (*n* = 93)	10.5 (9–11.5)		53 (57.0)	32 (34.4)	6 (6.5)	2 (2.1)	
4 children (*n* = 11)	10.5 (7.5–11)		7 (63.6)	3 (27.3)	0	1 (9.1)	
5 children (*n* = 1)	NA		0	1 (100)	0	0	

NA—not applicable.

**Table 3 vaccines-10-00404-t003:** Analysis between the knowledge score, knowledge rating and optional vaccination.

Knowledge Rating/Knowledge Score	Have You Given at Least One Optional Vaccine to at Least One of Your Children?		Statistical Analysis
	Yes, *n* (%)	No, *n* (%)	
Very good	1172 (60.5)	234 (38.2)	*p* < 0.001, χ^2^ = 93.9, OR = 2.5, 95% CI: 2.1–3.0
Good	636 (32.8)	274 (44.7)	*p* < 0.001, χ^2^ = 28.6, OR = 1.7, 95% CI: 1.4–2.0
Sufficient	120 (6.2)	91 (14.8)	*p* < 0.001, χ^2^ = 45.9, OR = 2.6, 95% CI: 2.0–3.5
Insufficient	9 (0.5)	14 (2.3)	*p* < 0.001, χ^2^ = 18.0, OR = 5.2, 95% CI: 2.2–12.0
Score	10 (IQR: 9–11.5)	9 (IQR: 7–10.5)	*p* < 0.001, U = 403203, z = −11.014, r = 0.22

**Table 4 vaccines-10-00404-t004:** Distribution of optional vaccines administered according to the child’s rank.

Type of Vaccine	Number of Vaccines Administered				Total, *n* (%)
	1st Child	2nd Child	3rd Child	4th Child	
Rotavirus vaccine	1165	500	41	2	1708 (28.1)
Varicella vaccine	945	293	28	1	1267 (20.9)
Pneumococcal conjugate vaccine	0	0	0	0	0 (0)
Meningococcal vaccine ACYW serogroups	460	125	14	1	600 (9.9)
Meningococcal vaccine B serogroup	316	91	11	0	418 (6.9)
Live attenuated influenza vaccine	354	107	8	0	469 (7.7)
Influenza inactivated vaccine	834	304	25	1	1164 (19.2)
Hepatitis A vaccine	286	86	8	0	380 (6.3)
I don’t know/I forgot	34	20	8	0	62 (1.0)
Total	4394	1526	143	6	6069 (100)

## Data Availability

The datasets generated and analyzed during the current study are available from the corresponding author (V.D.M.) upon reasonable request.

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
