# Peer review of "Optional Vaccines in Children—Knowledge, Attitudes, and Practices in Romanian Parents"

_vaccines, 2022, doi:10.3390/vaccines10030404_

Round 1

Reviewer 1 Report

The authors had provided good insight into the vaccine practice in the Romanian population. Although the study is about the survey for the vaccination record in different sections of society and there are minor English mistakes that can be corrected after the editor's decision.

Author Response

Thank you for your appreciation and recommendations. The manuscript has been revised for English. Several sentences have been corrected and reworded by an accredited English translator. These have been highlighted in the text with Track-changes. 

Reviewer 2 Report

The manuscript is well written with good details provided of patient histories and data presentations and analysis. There are a few suggestions to improve the quality of the manuscript as follows.

Text reference citations

Reference numbers throughout the manuscript should be indicated at full-size text font and in brackets [#] following the cited reference author(s) names or at the end of the sentence to give proper emphasis and indication of the information sources (making them more visible).

References

References 23 and 24 have the ref numbers incorrectly copied at the beginning of the reference line

Please provide DOIs for all references (for which they are available) 

Author Response

Thank you for your appreciation and recommendations. The manuscript has been revised for English. Several sentences have been corrected and reworded by an accredited English translator. These have been highlighted in the text with Track-changes. 

References have been added in brackets [x]. The list of references has been corrected and DOIs have been added. 
